# Female students' perspectives on cervical cancer screening inclusion in pre-admission university requirements in Ghana

**Evans Osei Appiah**[1,2]*, **Stella Appiah**[3], **Mary Ani-Amponsah**[4], **Boateng Susana Agyekum**[3], **Janet Acquah**[3], **Anna Nyamekye Addo Gyasi**[1], **Cindy Ofori-Appiah**[5]

**1** Department of Midwifery, School of Nursing and Midwifery, Valley View University, Oyibi, Ghana, **2** Purdue University, West Lafayette, IN, United States of America, **3** Department of Nursing, School of Nursing and Midwifery, Valley View University, Accra, Ghana, **4** Maternal and Child Health Department, School of Nursing and Midwifery, College of Health Sciences, University of Ghana, Accra, Ghana, **5** Ghana Christian University College, Amrahia, Ghana

\* oseiappiahevans@ymail.com

**Data Availability Statement:** All supporting data have been made available in the manuscript.

## Abstract

Cervical cancer is a huge contributor to cancer-related deaths and the commonest gynecological cancerous growth among women globally. Nevertheless, evidence suggests that cervical cancer incidence and mortality could be reduced through early diagnosis. Regardless of the accessibility of cervical cancer screening in Ghana, low reports of cervical screening had been recorded among female students and women in Ghana. The study objectives were to explore.Female students' opinions on the inclusion of cervical cancer screening in the pre-university admission screening requirement in Ghana. The facilitators and barriers to cervical cancer screening among female University students.Qualitative exploratory-descriptive design was employed for the purpose of this study. The target population was female students in a public university in Ghana who were purposively selected. Content analysis was employed for the data analysis. In all, 30 female students were selected to engage in face-face interviews using a semi-structured interview guide. Two categories and seven sub-categories were generated from the study analysis. It was interesting to know that majority 20 (66.66%) of the students supported adding CCS to the preadmission screening requirement with few refuting it. Others also recommended compulsory screening as a means of enhancing screening practices. The reasons for refusing this proposal by a few 10 (33.3%) of the participants were it being burdensome, time-consuming, and capital intensive. Other reasons for refusing it were sexual inactiveness following the screening, fear of discomfort, and the screening results. In conclusion, the study found that students were willing to undergo CCS if made mandatory for admission and suggested it be included in pre-admission screening requirements to encourage more Ghanaian women to participate. As CCS is effective in reducing cervical cancer incidence and burden, the proposal of including it in pre-university screening should be considered to increase uptake.

**Funding:** The authors received no specific funding for this work.

**Competing interests:** The authors have declared that no competing interests exist.

**Abbreviations:** ACS, American Cancer Society; CDC, Center for Disease Control; CC, Cervical Cancer; CCS, Cervical Cancer Screening; DHRCIIRB, Dodowa Health Research Center Institutional Review Board; HIV, Human Immuno Deficiency Virus; HPV, Human Papilloma Virus; VIA, Visual inspection with Acetic acid wash; WHO, World Health Organization.

## Introduction

Cervical cancer, representing 13% of female cancer growths and a huge contributor to cancer-related deaths, is the commonest gynaecological cancerous growth among women globally [1–3]. According the WHO, it is the most common cancer-causing mortalities among women globally with an estimated number of over 570 000 to 700000 cases between 2018 and 2030 [4]. Surprisingly, the authors revealed that a significant number of females affected are younger women. Nevertheless, there has been a projection that cervical cancer cases will decline by 42% by 2045, and by 97% by 2120.

About 85% of these cases occurred in technologically advanced countries. However, the burden is found to be higher in low- and middle-income countries than in advanced countries [5, 6]. Worldwide report in 2012 indicated that 266,000 women died of cervical cancer, which is equivalent to one woman dying every 2 minutes [7]. In sub-Saharan Africa, the mortality rate is accounted to be high (22.5) per 100,00 as compared to Western Asia, Western Europe, and Australia where the mortality rate is less [8]. American Cancer Society, therefore foresaw the diagnosis and deaths related to cervical cancer in America to increase to 12,360 and 4020 respectively. Cervical cancer was discovered to be more common among women with HIV especially those in Southern Africa and Eastern Africa with an estimation of more than 20 per 100 000 cases among inhabitants of these regions [9].

Fortunately, some authors have reported that cervical cancer could be prevented, and mortality be reduced through early diagnosis with cervical cancer screening [10–12]. In Ghana, a researcher supported this by indicating that cervical cancer can be eradicated through preventive screening and early detection of precancerous lesions [13]. A similar result was published in China that CCS can be utilized in the bid to dwindle the rate of cervical cancer load apart from the HPV antibodies and HPV DNA testing [14].

Despite the availability of cervical screening services in some hospitals in Ghana, less than 8.9% of the female students in Ghana reported having been screened for cervical cancer, even after being aware of the screening services [15]. Even though most studies have identified an appreciable level of knowledge on cervical cancer among female students the patronage rate in the screening among female students was reported to be low [16]. The major challenge is that most of the female students were engaged in factors that increase women's risk of CC such as pre-marital sex. Hence, some researchers in Ghana have suggested that University Health Services should develop strategies to help increase female students' interest in CCS [17].

At present, a national policy on cervical cancer screening is lacking in Ghana, and the existing screening guidelines are predominantly influenced by recommendations from institutions like the WHO [18]. Additionally, the authors noted that screening practices in Ghana are considered inadequate, with only 2.4% of the participants in a study having undergone a pap test. It was also discovered that the most prevalent Human papillomavirus (HPV) genotypes among women in Ghana were HR-HPV 16, 18, 45, 35, and 52 [19]. Although the cost of Pap tests has not been documented, the cost range for visual inspection with acetic acid (VIA) was reported to be between 4.93 US$ and 14.75 US$, while the cost of cryotherapy as a treatment option for cervical cancer ranged from 47.26 US$ to 84.48 US$ [20].

In Ethiopia, it was discovered that the knowledge about cervical cancer (95%), parity of women who had more children (95%), and perceived severity of the disease (95%) had a high correlation with the acceptance of cervical cancer screening [21]. Although a significant number of the respondents have heard of cervical cancer screening, approximately half (49%) did not know the cause, with about 71% being able to identify a few risk factors [22]. Strategies suggested to increase cervical cancer screening included increasing awareness, reducing screening costs, positive peer influence, and spousal support [23].

In Ghana, the cytology-based test is the most commonly utilized screening strategy [24]. Alongside this, the Visual Inspection with Acetic acid wash (VIA) has been utilized in both rural and urban settings in Ghana and is regarded as an effective screening method for cervical cancer and is available in almost every health care setting [25]. Regardless of the accessibility of these cervical cancer screening instruments, low reports of cervical screening had been recorded to be done for female students in Ghana and Uganda [26–29]. Moreover, some authors found out that three (0.8%) out of 392 participants had undergone CCS [21]. Furthermore, some researchers in Ghana realized that 12% of 140 female students had ever been screened for cervical cancer [22, 23]. Hence the researchers in this current study aim to explore the views of female students toward the inclusion of CCS in the preadmission screening requirement.

## Methodology

### Ethical statement

The Dodowa Health Research Centre Institutional Review Board (DHRC- IRB) provided clearance for this study to be conducted with the protocol number DHRCIIRB/073/05/21.

Verbal and written consent was sought from participants before the data collection.

### Research design

In this study, the qualitative exploratory-descriptive design was employed for the purpose of this study to fully understand the views regarding the addition of cervical cancer screening as part of the preadmission screening requirements among female students in a public university in Ghana. This design provides relevant preliminary insight into given problems, fills in gaps, and expands understanding as it explores research questions [30]. This design was selected because even though a lot is known about cervical cancer screening, there is scarce literature on students' opinions with regard to including cervical cancer screening in Preadmission screening requirements for university students. It also permitted the students to share their thought on why they think this proposal will work or not.

### Inclusion and exclusion criteria

The target population for this study was female students in a public university in Ghana. Participants included in this study were female students from 18 years since they have the capacity to give consent and those below 18 years who are emancipated adults, females of good physiological health as they are expected to be in the right mood and willing to take part in the interview; giving accurate information. Exempted from this study are male students and female students with mental illness.

**Sampling size and technique.** Purposive or judgmental sampling was used to select participants for the study. This ensured that only participants from the public university under study were selected from all levels and departments with the exception of those from health-related courses (nursing, medicine, pharmacy, and laboratory technician) based on the assumption that they might have learned a lot about cervical cancer screening which might influence their decision. This form of sampling technique allowed the researchers to select participants who qualify for the inclusion criteria and were willing to share their views. The guiding principle in sampling in qualitative studies is reaching data saturation. Thus, sampling ends at a point where no new information is obtained, and additional information does not modify your coding frame [31]. Therefore, the researcher interviewed participants till when no new information was retrieved which was on the 30th participant hence the sample size was 30.

A semi-structured interview guide designed by the researchers was used for the data collection and allowed probing into specific areas of interest to the researchers during the interviews.

**Data collection instrument and procedure.** The researchers sought an ethical clearance from the Dodowa Health Research Centre Institutional Review Board (DHRCIRB-DHRCIIRB/073/05/21). Following this, the study instrument was first pretested on two females at a private Christian university in Ghana helped the researchers to revise, remove and include other questions where appropriate which made the questions more concise and clearer. The data collection instrument was meticulously designed and reviewed by all authors based on relevant literature. The instrument comprised of three subsections. The first section sought to obtain demographic information about the participants. The second section solicited the perspectives of the students on the inclusion of cervical cancer screening in the pre-admission screening requirements. Finally, the last section gathered information on the facilitators and barriers to cervical cancer screening among the students. The data was collected by EOA and two other authors (SBA, JA, ANA), who received training on data collection from EOA, SA, and MA. These authors hold PhDs in Nursing, are nursing lecturers and advisors, and have extensive experience in conducting qualitative studies that have been published. Prior to the data collection, the purpose and procedure of the research were explained to the female students from the selected university who met the inclusion and were made to understand that they have the free will to decide whether to participate in the study. Following this, face-to-face in-depth interviews were conducted in English at a date and time suitable for the participants lasting for 3 months from August to October 2021. In all, thirty participants were interviewed. After the participants read the informed consent form, they were encouraged to seek clarifications to clear all misunderstandings and aided to append their signatures to the documents prior to the interview. During the interview, the researchers listened closely and reduced all forms of disruptions that might disrupt the conversation as they shared their thoughts about including CCS in the preadmission screening requirements. The researchers penned down issues some observed behaviors that could not be verbalized during the data collection for further clarifications.

The interviews were recorded using a voice recorder after seeking the consent from the participants. The researchers also recorded all field experiences and non-verbal communication (field notes) in their diaries. The entire interview lasted from fifty to sixty minutes. The participants were thanked for their time at the end of each interview. Member checking was done for further clarification.

## Data analysis

Data analysis is the "systematic organization and synthesis of research data and testing of research hypothesis using the data" [32]. The authors EOA, SA, MA, and CA, who possess comprehensive training in qualitative research from their Masters or PhD in Nursing, conducted the data analysis. Additionally, all authors were involved in verbatim transcription and familiarization and ensuring data accuracy. The researchers employed content analysis for the study analysis. The researcher transcribed the audio recorder verbatim in order to acquaint herself with the data. This was done through the process of familiarization, condensation, coding, categorization, and formulation of categories from the content. With familiarization, all the authors read through the transcripts severally to become familiar with the content and immerse themselves in the data collected this took the authors a week. Following this, the authors condensed participants' words by shortening them to a sentence that have a similar meaning to what the participants expressed. After this, the authors coded the content of the transcripts by labeling it with two to three words that have exact meaning to the condensed words. Categorization was done following coding by grouping together related codes and

further analysis produced sub-categories based on similarities and differences. Two categories and 7 subcategories were generated.

### Methodological rigour

The trustworthiness of the study was built based on four criteria proposed by Lincoln and Guba which were credibility, transferability, dependability, and confirmability [33]. The rigor was ensured by selecting participants who only met the inclusion criteria, piloting the data collection tool to ensure its reliability, and describing all the methodology subsections (Research design, data collection tool, data collection procedure, sampling technique, and size) to ensure confirmability and transferability, transcribing the recorded data verbatim and allowing other experts in the area to review the study to ensure that credible results are produced.

## Results

### Socio-demographic characteristics of participants

In this current study, thirty (30) participants were interviewed. The students were between the ages of eighteen (18) and twenty-five (25). All of them were females and stayed across the regions of Ghana. Twenty-two (22, 73.3%) were Christians, six (6, 20%) were Muslims, and two (6.6%) were non-Christians. They were all in their tertiary level and none had screened for cervical cancer. However, ten (33.3%) of the participants were sexually active with twenty (66.7%) non-active. the details are shown in Table 1:

### Organization of categories

During the analysis of the data, two (2) categories were generated. These categories were grouped into sub-categories and a total of seven (7) sub-categories were generated. Pseudonyms were used for each participant in order to maintain anonymity. See Table 2 for illustration of the categories and subcategories. A detailed description of the categories and sub-categories are presented in Table 2.

### Theme 1: Supporting the proposal of ccs inclusion in preadmission screening requirement

The participants of this recent study shared their thoughts about the inclusion of CCS into the preadmission screening requirement. Three (3) categories emerged from the analysis as:

1. Subscribing to the proposal

2. Mandatory screening

3. Willingness to screen to gain admission.

### Subscribing to the cervical cancer screening proposal

Majority of the current study participants (20) supported the idea that the addition of CCS to the preadmission requirements will be helpful to most females considering the rising numbers of cervical cancer in the country. This was explained in the quote below:

> 'Nowadays, I know that cervical cancer is rising and most of us have not been able to get screened even though some of we women have male sex partners. I believe some don't even know what the screening is and even if they know they don't care. So, I fully agree that the screening should be added to the requirements because when it happens like that most of the females will be screened for them to know whether they have cervical cancer or not.' P5, Age 20 years

**Table 1. Socio-demographic characteristics of respondents.**

| Variable | Frequency (n = 30) | Percent (%) |
|---|---|---|
| **Age group** | | |
| 18–20 | 9 | 30 |
| 21–22 | 7 | 23.3 |
| 23–25 | 14 | 46.7 |
| ≥26 | 0 | 0.00 |
| **Religion** | | |
| Christian | 22 | 73.3 |
| Muslim | 6 | 20 |
| Traditional | 2 | 6.6 |
| **History of CS screening** | | |
| Screened | 30 | 100 |
| Not screened | 0 | 0 |
| **Sexual history** | | |
| Active | 10 | 33.3 |
| Inactive | 20 | 66.7 |
| **Students' academic level** | | |
| 100 (Freshmen) | 15 | 50 |
| 200 (Sophomore) | 5 | 16.7 |
| 300 (Junior) | 5 | 16.7 |
| 400 (Senior) | 5 | 16.7 |
| Course | | |
| Business | 4 | 13 |
| Computer & Science and IT | 6 | 20 |
| Arts and Sociology | 8 | 27 |
| Chemistry and Physics | 8 | 27 |
| Political Science and Law | 4 | 13 |

Few participants 5 out of the 30 students explained that adding cervical cancer screening to preadmission screening requirement will help reduce cost which may increase the screening uptake.

**Table 2. Categories and sub-categories.**

| CATEGORIES | SUB-CATEGORIES |
|---|---|
| 1. SUPPORTING THE PROPOSAL OF CERVICAL SCREENING INCLUSION IN THE PREADMISSION SCREENING REQUIREMENT | 1. Subscribing to the cervical cancer screening proposal<br>2. Willingness to partake in CCS to gain admission.<br>3. Mandatory screening |
| 4. REFUTING CERVICAL CANCER SCREENING PROPOSAL | 1. Perceiving it as a burden, waste of time and money<br>2. Fear of results as a reason to refuse<br>3. Fear of sexual inactiveness after screening as a basis for refusal<br>4. Fear of discomfort as the basis for refusal |

Source: Interview Data, 2021

*'In my opinion I think it will be best if the add the cervical cancer screening to the screening requirements that we are to do before coming to school because I know that when we are many the price will be reduced by the hospital, I know that the screening is very expensive and I always hoped to do some but I do not have the means to, it was unfortunate I was not asked to do it before I was admitted.' P6 Age 18 years*

Another participant expressed interest in screening but outlined time factor as a hindrance to getting screened.

*'I have had an interest in going for the screening but my time doesn't allow me. I am student and my class schedules are a lot. Apart from studying I am also a caterer and I receive many orders so I fully support the idea of making students screen before been admitted it will help solve such challenges in the future.' P14 Age 23 years*

## Mandatory screening

Approximately, half of the participants 16 proposed that making the screening compulsory would facilitate CCS uptake in order to ensure early diagnosis and treatment.

*'I know that it is helpful, considering the increasing rate of the cancer. Just last week, a member of our church died from cancer and it was disheartening. So, personally, I believe that when it is made compulsory, every female student entering the university will have no option but to take it in order to gain admission.'P7 Age 19years*

It was reported that making CCS compulsory will enlighten them on the possible causes and erase any misperceptions that the student may have concerning CCS.

*'Not everybody knows about this screening so, I think that when it is made compulsory students every student entering the university will get to know of it and will help other females to do away with some misconceptions, they may have about the screening'P2, 18 years*

One participant saw no need to compel students to screen for cervical cancer as she viewed CCS as a personal choice that should be made by the individual.

*'I think that everyone has the right to choose what they feel is good for them and if they do not feel comfortable with the screening then their views should be respected because they may have other personal reasons related to why they do not want to get screened.'P14 23 years*

## Willingness to partake in ccs to gain admission

During the interviews, some participants expressed female students will be willing to partake in cervical cancer screening if added to the screening requirement in order to gain admission into the university.

*'At first, I don't want to do the screening because I don't see its importance as I am a virgin. I also find it uncomfortable to go naked in front of someone but I believe if they include it in the screening requirement, all the female students entering the universities will be willing to undergo the screening because they will want to be admitted.' P8, 22 years*

*"Every Ghanaian wants to be in the University now to get a good job, so if just going for cervical cancer screening will be one of the requirements to gain admission, I do not think it will be an issue for most females since everyone is capable of doing that unlike other requirements like academic performance"* P7 Age 19 years

Other participants also suggested that adding it to the university pre-admission screening requirement will make female students appreciate its importance to increase its acceptance.

*'I know that if it is included in the screening, it means it is something that is good, and hence female students will not have fear subscribing to it.'* P3, 21 years

It was revealed that there is a need for every female to have regular check-ups to be updated on their health conditions and getting screened was one of the important examinations that have to be considered.

*'As for me, I believe that every lady has to know about their health conditions and whatever goes on in her body. So, it is important to know about any infection that I may have before I marry, and sometimes, I worry that something may be wrong with me that I do not know of. I know how sexual activities can cause infections and I want to be sure that I am healthy. I have been wanting to have the screening so if it is part of the preadmission requirements of the school, but I did not get that opportunity because it was not a requirement* P21, 18 years

## Refuting cervical cancer screening proposal

Refuting the cervical cancer screening proposal as a preadmission screening requirement was the second theme that emerged from the study analysis. Some of the participants 10 out of the 30 students were against the notion of adding CCS to the preadmission requirements Four sub-categories were also generated under this theme which were Fear of results as a reason to refuse, Sexual inactiveness after screening as a basis of refusal and Fear of discomfort as a basis of refusal.

## Perceiving it as a burden, waste of time and money

Increase burden, busy schedules and financial constraints were the main reason why participants disagreed to the introduction of CCS as part of the preadmission screening requirement

*'I think it'll be a good idea for those who want to go for the screening. For me, it'll be a burden because I see it as a waste of time knowing that I not yet engaged in any sexual activity with any man. I am pretty sure that even if I do it, I will end up being hurt in the process because I heard they insert some things into the vagina.'* P9 Age 21 years

Herbal treatment served as the basis for showing reluctance towards the addition of CCS to the preadmission screening requirements by a participant.

*'My mum has always used herbs to treat my elder sisters and whenever we complain of any vaginal infection, it has helped us a lot. I don't see the need why I must go to the hospital to be screened and I do not see the need why they should add cervical cancer screening to other screening requirements prior to admission it will be a source of stress to other females who do not wish to undergo the screening like myself'* P15 Age 25 years

Financial constraint was revealed as the reason a participant was hesitant towards the proposal.

'*The fees here are very expensive and already, my parents are struggling to pay for my fees so I believe that when the screening is added to the preadmission requirements the cost will definitely raise and I won't be able to afford it.*' P11, 24 years

### Fear of results as a reason to refuse

Participants who knew the purpose of the CCS explained that they did not want to get screened because they were scared of what the results will be.

'*I had engaged in sexual activities when I was in SHS and now I have met someone who wants to marry me. Initially I was having sex with different people but then I heard that having sex with multiple people can cause cervical cancer so I stopped. So I'm not in support of this, what if I am tested positive? This is the main reason I think will scare other females too.*' P13, 19 years

'*Women are very emotional so we will not want to embark on something like this when we know the results will either be positive or I don't want any pressure so if I do it and I get an unexpected result I may not be able to handle the emotional trauma.*' P17, 20 years

Few participants indicated using herbal treatment as a means to escape anxiety accompanied by waiting for the results of the cervical cancer screening.

*.Some friends of mine went to do it and I could feel their anxiety as they waited for the results. Unfortunately, one had it, so I think they should leave us to make a decision whether to go for it all or not, after all, it is our life, I do not want to be waiting for any results which can give me unpleasant news.*' P15 Age 25 years

### Fear of sexual inactiveness after screening as a basis for refusal

Some participants alleged that CCS should not be included in the pre-university screening requirement because of the fear that it will make them sexually inactive or devirginize them.

'*I am a virgin and I fear and I want to keep my hymen intact I heard they will insert some tools into the vagina. I am not yet ready to get screened to loose what I have protected all these 21 years, They should let us be.*' P12, 21 years

'*I am afraid this procedure may interfere with a woman's sexual urge andability to enjoy sex like that why I think it is not necessary for them trying to force us to do this thing. I and my boyfriend are making plans of getting married soon so I don't want any complications from these insertions.*' P23, 25 years

Furthermore, another respondent recommended provision of education by the school on the effects of the screening on the sexual health of the individual.

'*Honestly, I am afraid I wouldn't be able to have sex after the screening. I don't even know if I would even feel like having sex. I haven't had much insight into the screening so I don't know lots of things about it. I think I'd have to read more about it after this interview. Women are inquisitive we only engage in things we know about, since we don't know about it, we will resist it even when it is included.*' P29, 24 years

### Fear of discomfort as the basis for refusal

Discomfort ensuing showing one's nakedness to another person along with the insertion of unknown tools into the cervix of the participants were labeled to serve as a deterrent to being screened.

*'Every woman is shy of her nakedness, especially me, I do not feel comfortable with other people seeing my nakedness. I have been trained never to let anyone see my nakedness apart from my husband, so the idea of someone else seeing my nakedness is very uncomfortable for me so I don't have an interest in getting screened at all. Left for me alone this proposal should never work.' P25, 18 years*

Mistrust about the sterility of the instruments to be used was pointed out by some participants.

*'The procedure is done by putting some equipment into the vagina and I don't know whether the tools that they use are always new. I do not trust the tools that they use because they may have been used on someone else so I am against this idea because I believe it will increase the number of sexually transmitted infections among women.' P26, 19 years*

Few participants were unwilling to be screened as they viewed the procedure to cause some sort of aches in their vagina.

*'I am sure that the process may be painful because I saw the instruments on the internet and it appears like the ones for surgery. And I belief it may be painful so I totally disagree to this.' P24, 22 years*

## Discussions

During the analysis of the study, it was realized that most of the participants (20) out of the 30 participants supported the motion for the inclusion of CCS in the pre university screening requirements, and 16 out of the 30 suggested including make it mandatory for all female students to partake in it. Even though previous studies have not looked at including CCS in university preadmission screening requirements, some studies have discovered the need for cervical cancer screening uptake its awareness and willingness to partake in it [23, 33, 34]. Due to the increasing number of women affected with cervical cancer, and the rise in mortalities associated with it, this could be an innovative way of increasing CCS uptake, helping in early diagnosis in order to prevent and initiate early treatment for those affected. This proposal could also aid to increase the awareness of cervical cancer screening among all female students. This was attributed to the fact that this will create awareness about CCS and entice the female students to be screened.

Another significant finding discovered was that if included in the Preuniversity screening requirement more women will be willing to screen in order to secure admission in the universities. This idea is supported by other studies where participants expressed willingness to screen to prevent CC and for early detection and treatment [35]. Gaining admission into the university has become competitive recently since most well-paid jobs require a university education for employment, and hence female students may not hesitate to screen even if they are unwilling to because they will not want to be denied the opportunity to gain admission to pursue a university education. Even though some of the students may not be doing it out of their free will, it could still go a long way to protect them from this condition and help them know their status early to prevent complications.

Few participants from the study 10 out of the 30 refuted the idea of adding CCS to the pre-admission requirements with the perspective that it is a waste of time, money, and burdensome. Similarly, other researchers have reported several barriers to cervical cancer screening including fear of exposing their nakedness, on the basis that it is costly, busy schedules, and fear of pain. This implies that these factors are still a challenge despite the advance in knowledge and technology [36, 37]. There is therefore the need for evaluation of programs instituted to improve CCS uptake and taking the perspectives of these women into consideration to help motive them to engage in the screening. It also implies that making including CCS in the pre-screening requirement at no cost, and making it accessible with no delays will enhance its acceptance and reduce the resistance.

Furthermore, other reasons for the refusal of this proposal were fear of results and sexual inactiveness following the screening. Akin to a present study, many students refused to participate in CCS due to the belief that the screening will take away their virginity, hence, saw no need to engage in the screening [17]. This brings to light that there is the need to increase education on how cervical cancer screening is done, the benefits, and some discomfort associated with it in order to reduce fear, clear misconceptions and increase acceptance of the screening. Women should be made aware that the benefits of the screening outweigh some of the discomforts and beliefs they have about it in order to motivate them to subscribe to the CCS. Fear prior to screening and following a test is expected however women should be made aware that going for the screening does not necessarily mean one will be tested positive nevertheless it helps one to be sure about her status and also help in detecting cancer at a stage when it could be treated. This fear in relation to cervical cancer screening results has been established by previous authors and as well as overcoming the fears to increase screening uptake [32, 37–40].

Unwillingness to be screened for cervical cancer was identified to stem from the discomfort that participants viewed could arise from the procedure which included showing one's nakedness to a stranger which paves the way for embarrassment, mistrust about the sterility of the instruments, and perceived vaginal pain due to the insertion of metallic objects into the intimate parts of the woman. Perceived vaginal pain was also identified among 9.4% of the participants in the study in Ghana, out of 140 females who believed that insertion of any objects into the reproductive parts of the female is bound to cause injury and consequently, pain [25]. This also adds to the fact that their knowledge of the CCS procedure was inadequate, hence, more education should be given on how the screening is performed. There should also be open forums where these students could ask questions for them to be addressed appropriately in order to facilitate increased CCS uptake.

## Conclusion

In conclusion, the study findings indicated that students were willing to undergo CCS if it was made mandatory for admission, and they recommended that the screening should be included in the pre-admission screening requirements to encourage more women in Ghana to participate. Evidently, CCS has been established as an effective method in reducing the incidence rate and burden of cervical cancer. Improving screening practices among female students will consequently improve the overall screening practices among women in the country. Therefore, it is recommended that the proposal of including CCS in the pre-university screening requirement should be given consideration to increase the uptake of the screening.

## Acknowledgments

The researchers wants to express their gratitude to all authors whose work were cited in this study the females students who took part in this study.

## Author Contributions

**Conceptualization:** Evans Osei Appiah, Stella Appiah, Boateng Susana Agyekum, Janet Acquah, Anna Nyamekye Addo Gyasi.

**Data curation:** Evans Osei Appiah, Stella Appiah, Boateng Susana Agyekum, Janet Acquah, Anna Nyamekye Addo Gyasi.

**Formal analysis:** Evans Osei Appiah, Mary Ani-Amponsah, Boateng Susana Agyekum, Janet Acquah, Anna Nyamekye Addo Gyasi.

**Investigation:** Evans Osei Appiah, Stella Appiah, Mary Ani-Amponsah, Janet Acquah, Anna Nyamekye Addo Gyasi, Cindy Ofori-Appiah.

**Methodology:** Evans Osei Appiah, Stella Appiah, Mary Ani-Amponsah, Janet Acquah, Anna Nyamekye Addo Gyasi, Cindy Ofori-Appiah.

**Resources:** Mary Ani-Amponsah, Janet Acquah, Anna Nyamekye Addo Gyasi, Cindy Ofori-Appiah.

**Software:** Evans Osei Appiah, Anna Nyamekye Addo Gyasi, Cindy Ofori-Appiah.

**Validation:** Evans Osei Appiah, Stella Appiah, Mary Ani-Amponsah, Cindy Ofori-Appiah.

**Visualization:** Evans Osei Appiah, Stella Appiah, Mary Ani-Amponsah, Cindy Ofori-Appiah.

**Writing – original draft:** Evans Osei Appiah, Stella Appiah, Mary Ani-Amponsah, Boateng Susana Agyekum, Janet Acquah, Anna Nyamekye Addo Gyasi.

**Writing – review & editing:** Evans Osei Appiah, Stella Appiah, Mary Ani-Amponsah, Boateng Susana Agyekum, Janet Acquah, Anna Nyamekye Addo Gyasi, Cindy Ofori-Appiah.

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
