## [Decision Letter · Decision Letter 0]

22 Sep 2022

PGPH-D-22-01164

FEMALE STUDENTS’ PERSPECTIVES ON CERVICAL CANCER SCREENING INCLUSION IN PRE-ADMISSION UNIVERSITY REQUIREMENTS IN GHANA

Dear Dr. Appiah,

Thank you for submitting your manuscript to PLOS Global Public Health. After careful consideration, we feel that it has merit but does not fully meet PLOS Global Public Health’s publication criteria as it currently stands. Therefore, we invite you to submit a revised version of the manuscript that addresses the points raised during the review process.

We look forward to receiving your revised manuscript.

Kind regards,

Vanessa Carels

Staff Editor

Journal Requirements:

1. Please change "female” or "male" to "woman” or "man" as appropriate, when used as a noun (see for instance https://apastyle.apa.org/style-grammar-guidelines/bias-free-language/gender.

2. Please provide separate figure files in .tif or .eps format.

Additional Editor Comments (if provided):

Reviewers' comments:

Reviewer's Responses to Questions

**Comments to the Author**

1. Does this manuscript meet PLOS Global Public Health’s publication criteria? Is the manuscript technically sound, and do the data support the conclusions? The manuscript must describe methodologically and ethically rigorous research with conclusions that are appropriately drawn based on the data presented.

Reviewer #1: Partly

2. Has the statistical analysis been performed appropriately and rigorously?

Reviewer #1: N/A

3. Have the authors made all data underlying the findings in their manuscript fully available (please refer to the Data Availability Statement at the start of the manuscript PDF file)?

Reviewer #1: Yes

4. Is the manuscript presented in an intelligible fashion and written in standard English?

Reviewer #1: No

5. Review Comments to the Author

Reviewer #1: Although the manuscript raises an interesting questions there are critical areas that calls for attention.

Objective: "explore the views of female students towards inclusion of CCS to the preadmission screening" requirement. This is too generic an objective with potential interpretations. What does view refers to when in specific during permission etc could have been specified as specific objectives.

Introduction: Given there are a lot of studies on CCS and probably dearth of evidence on students, the introduction section didn't provide convincing gaps that calls for this study. What other evidences are there in connection to students, admission etc is lacking from the introduction section.

Method - While CCS has been well studied, and given exploratory-descriptive design is meant to explore for the unknown, the choice of the design should have been justified.

Study participants are selected purposively but which class year and whee they have come from was not specified. This made it difficult to understand if the principle of maximum variation is taken in to consideration.

It is mentioned that "... choosing subjects who are believed to have fair knowledge about the content of the study and possess certain characteristics in which the researchers are interested in studying". What is fair knowledge and possession of certain characteristics should be clearer.

Saturation - I am sure saturation may not happen at the same number for all questions. However, while explaining this may help, the last number at which saturation happens should be indicated.

It is stated that "The study instrument was first pretested on two females at a private Christian university in

Ghana". I am wondering on why pretesting considered given there could be additional insights from the interview which may help sharpen the questions as you go along? any justification??. In fact you mentioned that "researchers penned down issues of interest in their diaries during the interview to aid in probing

for further clarification" which is an indication of an ongoing calibration of tool during data collection.

Data analysis is poorly presented including the choice of content analysis as a frame. Content analysis is about counting contents which is not the case from the findings and secondly themes and meanings were mentioned which is not aligned with content analysis. You may need to read again on content analysis and if you wish to maintain this for your analysis. Check the word crowding in a soft ware of your finding to present the finding interns of which contents appeared more. In this section, you have a title about rigour which in reality is about ethics. You need to explain what was done to ensure quality/rigor.

Result - The result section requires major revision to tighten the different sections of the findings. Critical is to have code book of themes, sub themes with explanations of what each stands for. Besides, it is critical to read through this section to refine the arguments. In fact there are shallow/crude words/statements in with such points as 'sexually active, both positive and negative views, and the theme themselves' that needs to be clarified and substantiated. Please avoid some, more, few ... which doesn't make sense unless you contextualise in the form of -- out ---. There are repetitions of findings that could be pulled under support, refute.... good examples are hesitation, reluctance, fear... which could be part of refuting. Please revise the themes to be retitled.

There are fallacies where willingness for screening was presented. While the students are already on campus their response on willingness for screening doesn't make sense. So, if the focuse is about pre-admission remain focused on that or if you want to measure both pre-admission and those already on campus then rework on your objective....

Discussion - Give there are some irregularities, crude statements, repetitions in the result section, the discussion section is not tight. What is it that this study could be known of? From this section it gives an impression that support and willingness themes are VERY IMPORTANT findings that is chosen for interpretation. Nonetheless, what was discussed about these remains too shallow and unwarranted. What is the outstanding finding from this study from the different theme or just one theme. How do you interpret this - what does this mean in view of available evidences? I think this part is too shallow.

6. PLOS authors have the option to publish the peer review history of their article (what does this mean?). If published, this will include your full peer review and any attached files.

**Do you want your identity to be public for this peer review?** For information about this choice, including consent withdrawal, please see our Privacy Policy.

Reviewer #1: **Yes: **Mirgissa Kaba

---

## [Decision Letter · Decision Letter 1]

17 Apr 2023

PGPH-D-22-01164R1

FEMALE STUDENTS’ PERSPECTIVES ON CERVICAL CANCER SCREENING INCLUSION IN PRE-ADMISSION UNIVERSITY REQUIREMENTS IN GHANA

Dear Dr. Appiah,

Thank you for submitting your manuscript to PLOS Global Public Health. After careful consideration, we feel that it has merit but does not fully meet PLOS Global Public Health’s publication criteria as it currently stands. Therefore, we invite you to submit a revised version of the manuscript that addresses the points raised during the review process.

Your manuscript has been evaluated by one new reviewer, and their comments are available below.

The reviewer has provided comments requesting additional details. In particular, please ensure that your stated conclusions are sufficiently supported by the findings of the study. Please provide a response to each of the reviewer's comments when revising your manuscript.

We look forward to receiving your revised manuscript.

Kind regards,

Hugh Cowley

Staff Editor

Journal Requirements:

Additional Editor Comments (if provided):

Reviewers' comments:

Reviewer's Responses to Questions

**Comments to the Author**

1. If the authors have adequately addressed your comments raised in a previous round of review and you feel that this manuscript is now acceptable for publication, you may indicate that here to bypass the “Comments to the Author” section, enter your conflict of interest statement in the “Confidential to Editor” section, and submit your "Accept" recommendation.

Reviewer #2: (No Response)

2. Does this manuscript meet PLOS Global Public Health’s publication criteria? Is the manuscript technically sound, and do the data support the conclusions? The manuscript must describe methodologically and ethically rigorous research with conclusions that are appropriately drawn based on the data presented.

Reviewer #2: Partly

3. Has the statistical analysis been performed appropriately and rigorously?

Reviewer #2: N/A

4. Have the authors made all data underlying the findings in their manuscript fully available (please refer to the Data Availability Statement at the start of the manuscript PDF file)?

Reviewer #2: Yes

5. Is the manuscript presented in an intelligible fashion and written in standard English?

Reviewer #2: No

6. Review Comments to the Author

Reviewer #2: Summary:

This qualitative study has tried to explore the perceptions of the female students about inclusion of cervical cancer screening as a pre-admission university requirement in Ghana. A total of 30 female students selected by purposive sampling were interviewed using semi structured interview guide.

General remarks

The authors have made tremendous effort to explore a very important research area (Cervical cancer screening) but at the same time, it is a well explored area with numerous publications, especially about barriers for cervical cancer screening. Though the authors have chosen a specific and interesting research question of opinion of students on inclusion of CCS in pre-admission university requirement, no new findings/barriers emerged from the study, apart from what is known from the previous literatures.

Comments

Introduction:

Needs to be more organized with details about cervical cancer screening guidelines followed in Ghana (Age recommendation for screening, whether done free of cost/paid, mandatory/optional) to make it more understandable for the international readers.

Clarify whether the recommendation for inclusion of CCS in pre-admission screening requirement was made by Government in Ghana or this study is based on recommendation made by authors of previous studies.

Methodology

Mention the gender, qualification and training in qualitative research of the person who collected the data

Provide details about the questions included in the semi-structured questionnaire

Data analysis

Mention how many authors were involved in the coding process and their training in qualitative research methods

The terms “categories and sub-categories” and “Themes and sub-themes” are used interchangeably by the authors in the data analysis and results section

Results:

While presenting socio-demographic characteristics of the participants, use either words or number to present the no. of participants in each category. Ex: Eighteen (18)-avoid using both.

In Table 1: Clarify what is this variable “Level” represents?

The results presented in Table 2 does not match with the Content analysis framework mentioned by the authors to have been used for analysis

Conclusion:

Conclusion provided by the authors “Undoubtedly, CCS has been proven effective in helping to reduce the incidence rate and the burden of cervical cancer” was not based on the findings of the present study

References:

There are errors in citing the references (Ex: 4,20,29). Check how to cite references for online resources, books etc

7. PLOS authors have the option to publish the peer review history of their article (what does this mean?). If published, this will include your full peer review and any attached files.

**Do you want your identity to be public for this peer review?** For information about this choice, including consent withdrawal, please see our Privacy Policy.

Reviewer #2: No

---

## [Decision Letter · Decision Letter 2]

26 May 2023

FEMALE STUDENTS’ PERSPECTIVES ON CERVICAL CANCER SCREENING INCLUSION IN PRE-ADMISSION UNIVERSITY REQUIREMENTS IN GHANA

PGPH-D-22-01164R2

Dear Mr. Appiah,

We are pleased to inform you that your manuscript 'FEMALE STUDENTS’ PERSPECTIVES ON CERVICAL CANCER SCREENING INCLUSION IN PRE-ADMISSION UNIVERSITY REQUIREMENTS IN GHANA' has been provisionally accepted for publication in PLOS Global Public Health.

Best regards,

Julia Robinson

Executive Editor

Reviewer Comments (if any, and for reference):

Reviewer's Responses to Questions

**Comments to the Author**

1. If the authors have adequately addressed your comments raised in a previous round of review and you feel that this manuscript is now acceptable for publication, you may indicate that here to bypass the “Comments to the Author” section, enter your conflict of interest statement in the “Confidential to Editor” section, and submit your "Accept" recommendation.

Reviewer #2: All comments have been addressed

2. Does this manuscript meet PLOS Global Public Health’s publication criteria? Is the manuscript technically sound, and do the data support the conclusions? The manuscript must describe methodologically and ethically rigorous research with conclusions that are appropriately drawn based on the data presented.

Reviewer #2: Yes

3. Has the statistical analysis been performed appropriately and rigorously?

Reviewer #2: Yes

4. Have the authors made all data underlying the findings in their manuscript fully available (please refer to the Data Availability Statement at the start of the manuscript PDF file)?

Reviewer #2: Yes

5. Is the manuscript presented in an intelligible fashion and written in standard English?

Reviewer #2: Yes

6. Review Comments to the Author

Reviewer #2: (No Response)

7. PLOS authors have the option to publish the peer review history of their article (what does this mean?). If published, this will include your full peer review and any attached files.

**Do you want your identity to be public for this peer review?** For information about this choice, including consent withdrawal, please see our Privacy Policy.

Reviewer #2: No
